# Distributed Energy Resource Exploitation through Co-Optimization of Power System and Data Centers with Uncertainties during Demand Response

Yu Weng [1,†] , Yang Liu [1,2,†], Rachel Li Ting Lim [1] and Hung D. Nguyen [1,*]

1   School of Electrical and Electronics Engineering, Nanyang Technological University,
    Singapore 639798, Singapore
2   Engie Lab, Singapore 118535, Singapore
*   Correspondence: hunghtd@ntu.edu.sg
†   These authors contributed equally to this work.

**Abstract:** This paper presents a robust bi-level co-optimization model that promotes the active participation of Internet Data Centers (IDCs) in demand response (DR) programs, thereby enhancing the flexibility of power systems. Our approach involves leveraging virtual power lines to migrate workloads among IDCs, optimizing resource allocations, and benefiting both domains. The model incorporates a Gaussian Process Regression (GPR)-constructed DR price–amount curve, which largely contributes to the simplification of the optimization problem with high accuracy and computational efficiency. It also respects the information barriers between the two domains of power systems and IDCs, and thus safeguards the privacy and flexibility of IDCs. The uncertainty in IDC operations is considered by incorporating the variance in GPR into the demand response curve. By integrating IDCs as DR resources, the framework of this research enhances the flexibility of power systems and the efficiency of cross-domain co-optimization. The model and algorithm are validated using modified IEEE test systems.

**Keywords:** bilevel optimization; demand response; Gaussian process regression; data centers; information barriers; price–amount curves; uncertainties





## 1. Introduction

With the auspicious development of computing technology and large language models, the global energy consumption of internet data centers (IDCs) reached 200 TWh in 2019, and its estimated electricity demand, together with the network service in 2030, may reach 4000 TWh, around 20.9% of the total global electricity consumption [1]. This huge energy demand will undoubtedly affect future power systems' operations and regulation [2], as such energy-intensive loads can alter the power flow patterns and incur safety risks [3]. Besides the high power demand, another special property of IDC is the flexibility to conduct geographical shifting, which can contribute to or harm the safe operation of power systems. Geographical shifting allows for the computing workloads to be migrated among different locations of IDCs [4,5], and also means that the cooperation between IDCs and power systems can achieve a win–win solution both economically and operationally.

In recent years, there has been an increasing focus on the cooperation between data centers (IDCs) and power systems to minimize operational costs [6]. The common approach for IDCs collaborating with power systems is by participating in demand response (DR) programs [7]. Reference [8] reviews the opportunities and challenges in this cooperation and provides directions for addressing these challenges. Paper [9] proposes a DR-pricing algorithm for IDCs that considers the uncertainties caused by prediction errors. In [10], a dynamic model of IDCs and their related subsystems is proposed, offering an accurate simulation of IDCs providing DR services. The dependencies between utilities and IDCs

are discussed in [11], which presents a DR pricing strategy based on game theory. The potential benefits of IDCs providing DR services are analyzed in [12], with simulation results showing up to a 40% reduction in IDC operational costs. The other literature highlights the benefits of IDCs providing DR services from different perspectives, such as fluctuation compensation [13] and virtual power plants [14].

However, applying the conventional demand response models directly to power systems with integrated IDCs leads to challenges [15]. These challenges arise due to the interests of cloud service providers (CSP) [16], the time-varying nature of computing workloads affecting DR availability [17], and the uncertainties caused by predicting interactive task flows and local generator outputs [18]. IDCs operate outside the purview of power system supervision and management [19]; thus, the leader–follower relationship employed in many bi-level optimization models does not hold [20]. Therefore, optimization models for co-optimization must acknowledge the unavailability of information within IDCs to power systems. Addressing these challenges requires a systematic solution that respects IDC's interests and privacy while ensuring computational efficiency [19,21].

In this paper, we propose a robust bi-level co-optimization model that explores the mechanism for motivating IDCs to provide their flexible loads as demand response (DR) resources to power systems. Our approach involves migrating workloads among IDCs through virtual power lines to enhance their participation. The model is based on a Gaussian process regression (GPR)-constructed DR price–amount curve [19,22], which respects information barriers, safeguarding the privacy and flexibility of IDCs. Uncertainties in IDCs' operations are incorporated into the demand response curve through the variance in GPR. We validate the effectiveness of our proposed model and algorithm using modified IEEE test systems with IDCs. Our main contributions are as follows:

- This paper proposes a two-layer robust optimization model to promote the active participation of IDCs in demand response, leveraging spatial migration with time-varying workloads among IDCs to optimize resource allocations and benefit both domains. Our cooperative mode between power systems and IDCs considers uncertainties on both sides, ensuring robustness.
- Data-based price–amount curves are constructed to bridge the communication between power systems and IDCs, which facilitates effective co-optimization and also protects the privacy of IDCs' information. The constructed curves can largely improve computational efficiency and avoid unwilling data exchange, which could be crucial in the near future as the number of IDCs continues to rise.
- Gaussian process regression is employed to construct price–amount curves by capturing IDC's behaviors following the change in price in the DR scheme with uncertainties. The GPR-constructed curves thus have explicit function forms with high accuracy, which can simplify the original two-layer problem while still allowing for desired adjustments in DR amount and enabling foresight regarding the uncertain availability of IDCs in the co-optimization.

## 2. Robust Co-Optimization Model of Power Systems and IDCs for Demand Response

As a future trend, more renewable energies will be integrated and form microgrid communities, and power systems may offer different electricity prices and DR policies to different communities; therefore, an economical operation is very necessary to maximize the revenue of IDCs.

In the following subsections, the power system operation and the IDC operation are modeled and explained, followed by a description of the co-operation methods of the two systems. It is worth noting that the DR service here refers to the mainstream definition of DR, where the load side voluntarily curtails its demand at a certain time [23], and the following models are expressed accordingly.

### 2.1. Interaction between Power Systems and IDCs

Following the auspicious development of IDCs, the high power demand and the virtual power lines among IDCs may change the power flow directions and the operation modes of power systems. In turn, the cap of the power supply also affects the operation of IDCs, such as the workload migration and termination. The details of this interaction are illustrated in Figure 1, wherein the two systems operate with their own objectives. The operation of each system has a significant impact on the key factors of the other. Consequently, operating the two systems independently may result in mutual disturbance, reducing efficiency and security. Conversely, when the systems are co-operated, a win–win solution can be achieved, where one system compensates for the shortcomings of the other. A prime example of this is the cooperation between IDCs and power systems, where IDCs provide demand response (DR) to earn additional profits, while the provided DR assists in maintaining power balances.

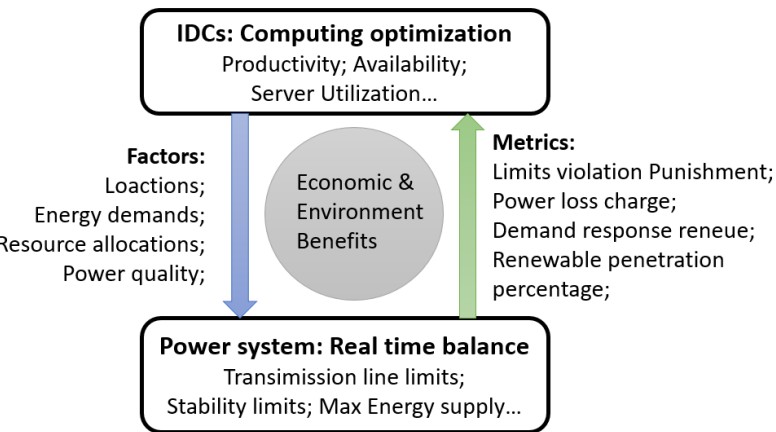

**Figure 1.** Interdependence between IDCs and power system.

A virtual power line can be defined as the virtual line connecting any two nodes at which the data centers are located. Virtual power lines could also form a meshed grid, which is different from the power grid if there are more than two data centers. Note that the amount of power change that occurs through virtual power lines is not the same as the actual power change in the network, for the following reasons. The first difference is in the power loss after the power demand's geographic shifting. The second difference is in the distributed renewable's contribution at locations. The third factor is the operational limits of the distribution network as the infrastructure in the distribution network is less robust compared to the transmission networks and also faces aging issues. Also, future flexible operation modes, like peer-to-peer energy trading, can largely alter the power flow patterns. Thus, due to the high nonlinearity of power systems, the effects of the power exchange on virtual power lines on the actual power network need to be further investigated for the safety of future power systems' operation and the co-optimization of data centers. Future works focusing on the difference between the power transfer on the virtual power grid and the actual power flow in the physical network will be introduced separately through machine learning methods. In this work, we limit ourselves to the two-layer co-optimization problem.

The interaction between power systems and IDCs can be represented by a general bilevel optimization framework, as shown in Figure 2, where the upper layer and the lower layer are mutually affected by the adjustable power demand and float electricity costs, while the autonomy and privacy of the lower layer are respected through a proposed regressed function to decouple the two layers in the bilevel problem and reduce the computational size and complexity. The proposed method provides a general solution for such bilevel problems in power systems with an explicit function to accurately reflect the behaviours of the IDCs.

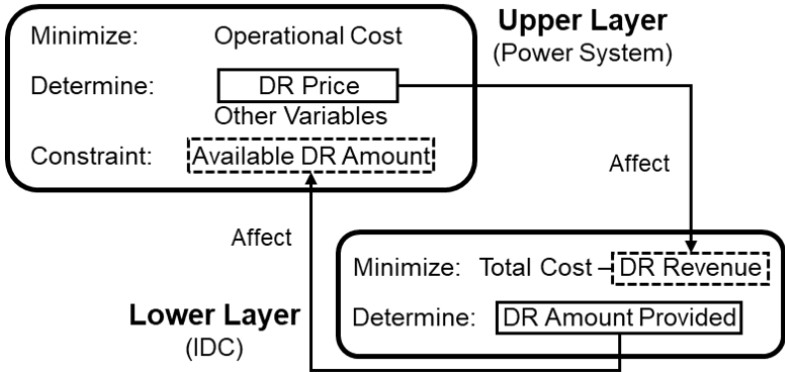

**Figure 2.** Bilevel Structure of the Interaction Between Power Systems and IDCs.

A general bilevel optimization framework can be described with the functions as follows. Power system model $\mathcal{M}_\mathcal{P}$:

$$\min_{x_\mathcal{P}} \quad c(x_\mathcal{P})$$
$$\text{subject to} \quad g(x_\mathcal{P}) \leq 0. \tag{1}$$

IDCs model $\mathcal{M}_\mathcal{D}$:

$$\max_{x_\mathcal{D}} \quad C(x_\mathcal{D})$$
$$\text{subject to} \quad G(x_\mathcal{D}) \leq 0. \tag{2}$$

Here, $x_\mathcal{P}$ and $x_\mathcal{D}$ denote the variables in power systems relevant to its optimal operation and those in IDCs. As IDCs and power systems are coupled, we have that the coupling variable set $x_{\mathcal{D}\mathcal{P}} = x_\mathcal{D} \cap x_\mathcal{P}$ is a non-empty set. The coupling between IDCs and power systems will be reflected by the interactions between the two models $\mathcal{M}_\mathcal{D}$ and $\mathcal{M}_\mathcal{P}$. In this paper, the interaction between power systems and IDCs is described as below.

$$\min_{\mathbf{X}^{\mathbf{ps}}} C(\mathbf{X}^{\mathbf{ps}}, DR^{IDC}) \tag{3}$$

subject to

$$DR^{IDC} \in \underset{\mathbf{X}^{\mathbf{IDC}}}{\arg\min} \{c(p^{DR}, \mathbf{X}^{\mathbf{IDC}}) : g(p^{DR}, \mathbf{X}^{\mathbf{IDC}}) \leq 0\} \tag{4}$$

$$p^{DR} \in \mathbf{X}^{\mathbf{ps}} \tag{5}$$

$$G(\mathbf{X}^{\mathbf{ps}}, DR^{IDC}) \leq 0 \tag{6}$$

$C$ and $c$ represent the objective function of power systems and IDCs, respectively; $G$ and $g$ are the operational constraints of power systems and IDCs; $\mathbf{X}^{\mathbf{ps}}$ and $\mathbf{X}^{\mathbf{IDC}}$ represents the operational variables of power systems and IDCs. The upper layer represents power systems, and the main decision variable is the DR price. The determined DR price will affect the DR revenue of IDCs, and further affect the DR amount provided by IDC in the lower layer. For IDCs, providing DR to power systems brings additional revenue, which partially offsets the operational cost. If the IDC is assumed to be a price taker [6,24], the amount of DR provided IDC is affected by the DR price, which is determined by power systems. Therefore, power systems should determine the DR price according to the required DR amount, and IDCs will decide the actual DR amount according to the DR price.

### 2.2. Power System Operation Model

The power system operation model with uncertain renewable energy is introduced in this subsection. The aim of the model is the minimization of total operational cost to achieve a power balance between purchased DR and reserve capacity.

$$\min\{C^G + C^{DR} + C^R\} \tag{7}$$

where $C^G = \sum_{i,t}\{a_i^G * PG_{i,t}^{0\,2} + b_i^G * PG_{i,t}^0\}$, $C^R = \sum_{i,t}\{a_i^R * R_{i,t}^{up} + b_i^R * R_{i,t}^{down}\}$, $C^{DR} = \sum_{i,t} p_{i,t}^{DR} * DR_{i,t}$. The first item in the objective function describes the generation cost expressed as a quadratic form, and the operational cost of renewable units is assumed to be zero. The second item is the generation reserve operational costs, where different cost coefficients are assigned to upward and downward reserves. The third item describes the DR purchase cost. The constraints of power system operation model are given as Equations (8)–(17).

$$\sum_j B_{i,j}(\theta_{i,t}^0 - \theta_{j,t}^0) = PG_{i,t}^0 - PD_{i,t}^0 \qquad \forall i,t \tag{8}$$

$$\sum_j B_{i,j}(\theta_{i,t} - \theta_{j,t}) = PG_{i,t} - PD_{i,t} \qquad \forall i,t \tag{9}$$

$$PD_{i,t}^0 - DR_{i,t} \le PD_{i,t} \le PD_{i,t}^0 \qquad \forall i,t \tag{10}$$

$$PG_{i,t}^0 - R_{i,t}^{down} \le PG_{i,t} \le PG_{i,t}^0 + R_{i,t}^{up} \qquad \forall i,t \tag{11}$$

$$0 \le R_{i,t}^{up} \le r_{i,t}^{up} \qquad \forall i,t \tag{12}$$

$$0 \le R_{i,t}^{down} \le r_{i,t}^{down} \qquad \forall i,t \tag{13}$$

$$\underline{PG_i} \le PG_{i,t}^0 \le \overline{PG_i} \qquad \forall i,t \tag{14}$$

$$\underline{PG_i} \le PG_{i,t} \le \overline{PG_i} \qquad \forall i,t \tag{15}$$

$$-P_{i,j}^{Line} \le B_{i,j}(\theta_{i,t}^0 - \theta_{j,t}^0) \le P_{i,j}^{Line} \qquad \forall i,j,t \tag{16}$$

$$-P_{i,j}^{Line} \le B_{i,j}(\theta_{i,t} - \theta_{j,t}) \le P_{i,j}^{Line} \qquad \forall i,j,t \tag{17}$$

$$\underline{\theta_i} \le \theta_{i,t}^0 \le \overline{\theta_i} \qquad \forall i,t \tag{18}$$

$$\underline{\theta_i} \le \theta_{i,t} \le \overline{\theta_i} \qquad \forall i,t \tag{19}$$

Since the generation output of renewable units is uncertain but predictable [25], the power balance should first be achieved by scheduling the conventional units and the DR when the renewable generation is perfectly predicted (Equation (8)). Also, sufficient reserve capacity and DR resources should be prepared to cover the possible deviations between the actual output and the predicted value of renewable units (Equations (9)–(11)). The reserve capacity should be within the generators' ramping limits (Equations (12) and (13)). The scheduled generation and the actual generation of generators should be within the generators' output limitation (Equations (14) and (15)). Equations (16) and (17) describe the power cable transmission capacity, and the voltage angle limits are modeled in Equations (18) and (19). It is worth noting that the uncertainty of load demand forecasting in power system operation is not considered here, and the nominal load amount obtained from historical data is used to represent the load demand, as shown in Equation (20). To simplify the model, for the buses with IDCs ($\mathcal{B}^{idc}$), it is assumed there is only one IDC in each bus (the IDC is expressed as $idc^{\{i\}}$), and the IDC is the only load in that bus. Hence, the DR resources in these buses are all provided by IDCs (Equation (21)).

$$PD_{i,t}^0 = PD_{i,t}^{Nomi} \qquad \forall i,t \tag{20}$$

$$DR_{i,t} = DR_{idc^{\{i\}},t}^{IDC} \qquad \forall i \in \mathcal{B}^{idc}, t \tag{21}$$

## 2.3. Data Center Operation Model with Adjustable Loads

The operation of IDCs equipped with renewable local generators and ESS is modeled in this subsection. The robust optimization, which is widely applied in IDC operations [7], is used here to minimize the operational cost in objective function (22) wherein considers the

uncertainties coming from renewable, incoming computing tasks, and predicted electricity prices:

$$\min_{\mathbf{V^{1st}}} \left\{ \begin{array}{l} \displaystyle\sum_{wl} v_{wl}^t \, p_{wl}^{WL} \\[2mm] + \displaystyle\max_{\mathbf{V^{2nd}}} \left\{ \displaystyle\sum_{idc,t} PD_{idc,t}^{IDC} \, p_{idc,t}^e - \displaystyle\sum_{idc,t} DR_{idc\{i\},t}^{IDC} \, p_{idc,t}^{DR} \right. \\[4mm] \hspace{5cm} \left. : \mathbf{Z} \in [\underline{\mathbf{Z}}, \overline{\mathbf{Z}}] \right\} \end{array} \right\} \tag{22}$$

where $\mathbf{V^{1st}}$ is the first-stage variable set describing the workload adjustment in IDCs; $\mathbf{V^{2nd}}$ is the second-stage variable set including $DR^{IDC}$ and the other power-balancing variables in IDCs. The value of $\mathbf{V^{2nd}}$ will be optimized after the uncertainties are known; $\mathbf{Z}$ is the uncertain parameter set; $\overline{\mathbf{Z}}$ and $\underline{\mathbf{Z}}$ describe the varying range of $\mathbf{Z}$. The first item of Equation (22) is the workload termination cost. These workloads are scheduled to be completed in the operation periods, but are terminated to reduce power demand and provide DR to power systems. The second item is the electricity bill cost. The third item is the revenue obtained by providing DR to power systems, which $p_{idc\{i\}}^{DR}$ means that the DR price of bus $i$ is where the IDC is located.

The power demand of a IDC consists of two parts: the IT equipment power demand and the cooling demand. It is generally believed that the cooling demand is proportional to the IT demand. Therefore, a power usage effectiveness (*PUE*) index is used to describe the ratio between the total power demand (IT + Cooling) and the IT demand. In addition to the ESS charging/discharging demand and local generator output, the total IDC power demand is expressed in Equation (23). The power demand of IDC should be less than or equal to the difference between nominal demand and the provided DR amount. (Equation (24)). Considering that the output of local renewable generator $PG^{IDC}$ and the IDC power demand $PD^{IDC}$ are uncertain, the difference between the IDC nominal demand and the actual demand is treated as the demand response provided by IDCs after the uncertainties are known.

$$PD_{idc,t}^{IDC} = PD_{idc,t}^{IT} * PUE_{idc,t} + P_{idc,t}^{ESS} - PG_{idc,t}^{IDC} \tag{23}$$

$$PD_{idc,t}^{IDC} \leq P_{idc,t}^{Nomi} - DR_{idc,t}^{IDC} \tag{24}$$

$$E_{idc,t}^{ESS} = E_{idc,t}^{In} + \sum_{t'=1}^{t} P_{idc,t'}^{ESS} \tag{25}$$

$$0 \leq E_{idc,t}^{ESS} \leq E_{idc,t}^{Max} \tag{26}$$

where all constraints hold $\forall idc, t$.

Two types of IDC workload are considered in this paper: the interactive ones and the flexible ones. The required server amount for an interactive workload is not known before it arrives (but can be forecasted), and the required server has to be satisfied in real-time. For a flexible workload, the required server amount is known, and the workload can be migrated among IDCs and time slots, or terminated when necessary. IDCS should sufficiently allocate servers to provide for the predicted workload. It should also migrate or terminate flexible workloads according to the available server amount.

$$S_{idc,t}^{usage} = S_{idc,t}^{in} + \sum_{wl} S_{wl,idc,t} \hspace{3cm} \forall idc, t \tag{27}$$

$$0 \leq S_{idc,t}^{usage} \leq S_{idc}^{cap} \hspace{3.8cm} \forall idc, t \tag{28}$$

$$PD_{idc,t}^{IT} = S_{idc,t}^{usage} * \rho \hspace{3.5cm} \forall idc, t \tag{29}$$

The server allocation is modelled in Equations (27) and (28). Based on the allocated server amount, the IT equipment power demand can be calculated, as shown in Equation (29), where $\rho$ is the coefficient describing the power demand of one unit of allocated computa-

tional resource [16]. Since the interactive workloads $S^{in}$ are uncertain, the IDC IT demand $PD^{IT}$ and the actual demand $PD^{IDC}$ are both uncertain.

$$v_{wl,idc}^c \geq \frac{1}{M}\left(\sum_{t\in[\underline{T}_{wl},\overline{T}_{wl}]} S_{wl,idc,t} - S_{wl}^{req} + m\right) \qquad \forall wl, idc \qquad (30)$$

$$v_{wl,idc}^c \leq 1 - \frac{1}{M}\left(S_{wl}^{req} - \sum_{t\in[\underline{T}_{wl},\overline{T}_{wl}]} S_{wl,idc,t}\right) \qquad \forall wl, idc \qquad (31)$$

$$\sum_{idc} v_{wl,idc}^c \leq 1 \qquad \forall idc \qquad (32)$$

$$v_{wl}^t = 1 - \sum_{idc} v_{wl,idc}^c \qquad \forall wl \qquad (33)$$

Equations (30)–(33) describe the flexible workload's completion. A flexible workload will be completed in an IDC ($v_{wl,idc}^c$ equals 1) when the total allocated resource amount between its release time and deadline equals the required amount (Equations (30) and (31)). Equation (32) shows that each workload can only be completed once, and the uncompleted workload will be terminated ($v_{wl}^t$ equals to 1), as shown in Equation (33).

$$v_{wl,idc,t}^u \leq v_{wl,idc}^l \qquad \forall wl, idc, t \qquad (34)$$

$$S_{wl,idc,t} \leq \left(v_{wl,idc}^l + \sum_{idc}\sum_{t'\in[1,t-T^d]} v_{wl,idc,t'}^u\right) M \quad \forall wl, t \qquad (35)$$

$$\sum_{wl} v_{wl,idc,t}^u \leq up^{max} \qquad \forall idc, t \qquad (36)$$

Equations (34)–(36) describe the flexible workload migration along the virtual power lines among IDCs, where $v^l$ is the binary parameter indicating the original location of a flexible workload (1 if a workload is located in the IDC). Equation (34) shows that a workload can only be calculated in its original location before it is uploaded to the cloud after a certain time delay $T^d$, and Equation (35) indicates that a workload can only be uploaded from its original location. The uploading limit of workloads is described in Equation (36).

## 3. Solving the Co-Optimization Problem with a GPR-Based DR Price–Amount Curve

This section introduces the Gaussian process regression methods to construct price–amount curves to facilitate effective co-optimization and preserve the privacy of IDCs' inner models and information. An explicit function of the price–DR amount curve with high accuracy can be obtained from variables $p^{IDC}$ and $DR^{IDC}$ as follows:

$$DR_{idc\{i\},t}^{IDC} = f(p_{i,t}^{DR}) \qquad (37)$$

Equation (37) essentially describes a DR price–amount bidding curve, which should be submitted by DR suppliers in DR programs [23]. For conventional DR suppliers, the value of their load does not vary over time. Therefore, the DR bidding curve can be empirically generated based on historical data, as shown in Figure 3. However, this conventional way is not suitable for IDCs because the workloads in IDCs can be migrated temporally and spatially, and the uncertainties in IDC operation make it more difficult to obtain the DR bidding curve. An alternative for IDCs is to directly estimate Equation (37) based on the predicted IDCs' operations, and then submit the estimated function to the power system as the DR bidding curve. In this paper, Gaussian process regression is utilized to acquire this function.

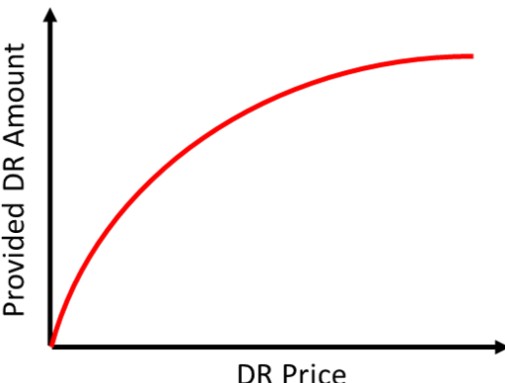

**Figure 3.** Conventional DR Price-Amount Curve.

*3.1. Gaussian Process Regression*

To regress a function $y = f(x)$, a sample set $S$ is needed, including the controlled variable $D = [x^{(0)}, x^{(1)}, ..., x^{(m)}]$ and the dependent variable $R = [\hat{y}^{(0)}, \hat{y}^{(1)}, ..., \hat{y}^{(m)}]$. With GPR, the regressed form of the function will be:

$$\hat{y} = \hat{f}(x) = k(x, D)^T (k(D, D) + \sigma^2 I)^{-1} R \qquad (38)$$

GPR also considers the uncertainties in the original function and the noise in the sampling process, and calculate the varying range of the regressed function as:

$$\Sigma(x) = k(x, x) + \sigma^2 I - k(x, D)(k(D, D) + \sigma^2 I)^{-1} k(D, x) \qquad (39)$$

where $k$ is a kernel function.

Compared to other, similar methods, GPR has the following advantages [19,22]:

(1)　An analytic expression instead of a black box is used to describe the function, which means that the regressed function can be directly used on the power system side;
(2)　As a non-parametric method and does not require certain forms of the regressed function;
(3)　As a Bayesian method, GPR can calculate the varying range of the estimated function;
(4)　Despite the limited number of samples, an accurate estimation can still be achieved with proper kernel selection.

*3.2. Procedures of DR Price–Amount Curve Construction with GPR*

The procedures used to estimate Equation (37) for the DR price–amount curve, as demonstrated in Figure 4, can be summarized as follows:

1.　Generate sample set. Randomly sample control variable $p^{DR}$; calculate the related dependent variable $DR^{IDC}$.
2.　Calculate the first-stage variables $\mathbf{V^{1st}}$ by solving the IDC optimization model (Equations (22)–(36)) using calculated $DR^{IDC}$.
3.　With the obtained $\mathbf{V^{1st}}$, vary the range of the uncertain parameters $\mathbf{Z}$ according to its probability distribution. Solve IDC optimization model (Equations (22)–(36)) again.
4.　Record the data. Repeat the above steps until we obtain a set of data for DR price and demand $D = [p^{DR(1)}, p^{DR(2)}, ..., p^{DR(m)}]$ and $R = [DR^{IDC(1)}, DR^{IDC(2)}, ..., DR^{IDC(m)}]$.
5.　Estimate the DR price–amount bidding curve with data set $D$ and $R$ using GPR with Equations (38) and (39). Acquire function $DR^{IDC} = \hat{f}(p^{DR})$ and the variance in the curve $\Sigma(p^{DR})$.
6.　Take function $DR^{IDC} = \hat{f}(p^{DR})$ in the power system operation model Equations (7)–(21), the optimal DR price can be calculated.

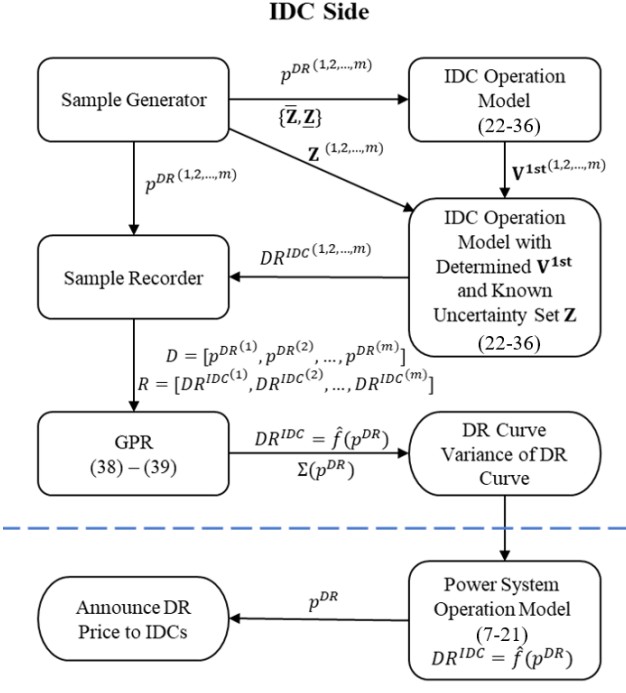

**Figure 4.** Procedures Illustration of the Proposed Algorithm.

The major difference between the proposed GP-based method and the conventional methods is the lower-layer models and constraints. As a common practice, in the conventional methods, the solution is found by integrating IDC models (i.e., Equations (22)–(36)) and power system models (i.e., Equations (8)–(21)) into a single-level model. iI the proposed GP-based model, it is only necessary to solve the upper-layer model with a regressed function $DR^{IDC} = \hat{f}(p^{DR})$, which represents the lower-layer models, so that the total computational cost is decided by the upper-layer model.

In other words, the lower-layer model in Equations (22)–(36) is replaced by the function $\hat{f}(\cdot)$. This function largely improves computational efficiency and avoids unwilling data exchange.

## 4. Case Studies

### 4.1. Optimal Operation of Microgrids with Fast Response Units and IDCs

The microgrid network topology for the case study is shown in Figure 5. The microgrid is revised based on the standard IEEE 15-Bus system in Matpower "case15da". The voltage base is 11 kV and the power base is 1 MVA. The demand in the system is powered by the substation and two wind units. The system operator can purchase reserve capacity from the two fast response (FR) units, or DR resources from the two IDCs. Table 1 shows the detailed system information. For each IDC, the nominal demand is 100 MW, and the scheduled workload number at each time slot is 60. Local PV units are installed in the IDCs, which brings uncertainties to the DR provided by IDCs.

The proposed method is applied to determine the DR price and the related reserve capacity purchase amount. According to the procedures in Figure 4, the first step is to estimate the function (Equation (37)) between the DR price and the DR amount provided by the IDCs. Because Equation (37) contains multiple dimensions, it is difficult to visualize the function as the curve shown in Figure 3. However, we can still calculate the accuracy of the estimated function to see whether it can accurately reflect the change in DR amount with DR price. After the DR price–amount curve is estimated, the optimal DR price and the related DR amount provided by IDCs can be calculated.

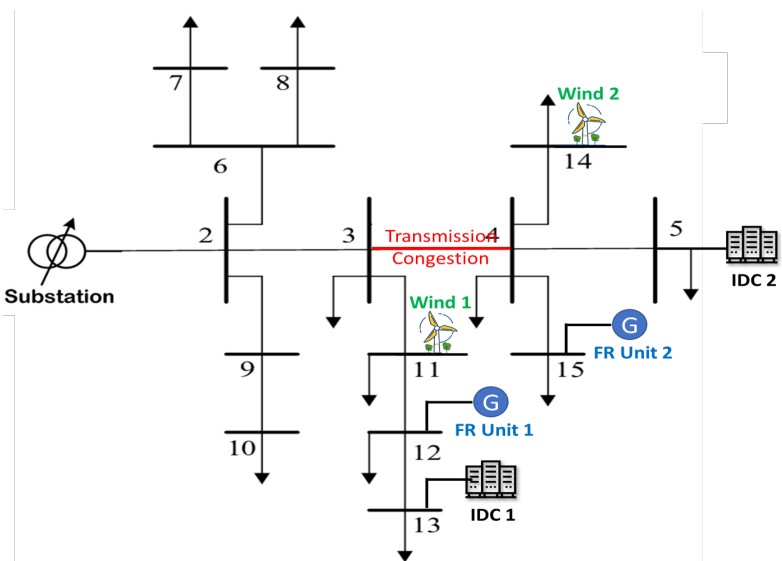

**Figure 5.** Microgrid with Renewable Units, Fast Response Units and IDCs.

**Table 1.** Modified IEEE 15 Bus System Information.

| Fast Response Unit | Reserve Price ($) | Wind Unit | Predicted Output (MW) | Worst Scenario Output (MW) |
| --- | --- | --- | --- | --- |
| FR Unit 1 | 150 | Wind 1 | 200 | 50 |
| FR Unit 2 | 200 | Wind 2 | 150 | 0 |
| IDC | Nominal Demand (MW) | Workload Number | Local PV Predicted Output (MW) | Local PV Standard Deivation |
| IDC 1 | 100 | 60 | 20 | 7 |
| IDC 2 | 100 | 60 | 20 | 7 |

The DR and reserve purchase scheme based on the proposed method is compared to the optimal solution derived from the global searching in the feasible regions of the original bi-level problem, and the results are shown in Table 2. The outcome of the proposed method is close to the optimal solution, proving the effectiveness of the proposed method.

**Table 2.** DR and Reserve Purchase Results.

| | DR Cost ($) | Reserve Cost ($) | IDC1 DR Price ($) | IDC2 DR Price ($) |
| --- | --- | --- | --- | --- |
| Proposed Method | 6.89k | 21.67k | 33.6 | 44.5 |
| Global Optimal | 6.65k | 22.63k | 35 | 43 |

*4.2. Transmission Line Congestion and Virtual Power Line*

This section shows a case study focusing on how the workload migration between IDCs helps with power network congestion and also reduces the economic costs of resource purchases. The concept and the simulation results are presented in Figures 6–9.

In the test cases, the congestion occurs when the current flow in the line is higher than 1.5 times the nominal current in Matpower's standard case. Transmission lines 3–4 were selected to demonstrate the operational and economic benefits of workloads migration. The concept is illustrated in Figure 6. As shown in Figure 6, Zone 1 contains the renewable 1, IDC 1 and the FR unit 1, while Zone 2 has renewable 2, IDC 2 and the FR unit 2. The selected lines 3–4 bridge Zone 1 and Zone 2. However, the demand response and the renewable power in Zone 1 cannot be used to balance the power shortage or deviation in Zone 2 if congestion happens on lines 3–4. Congestion could happen when any operational

constraints are violated, including voltage, current, thermal, power, and stability constraints. In other words, any power flows across zone 1 and zone 2 could aggravate the congestion on lines 3–4, as indicated by Flow 1 in Figure 6. Therefore, by taking advantage of the workload migration of IDCs, the power can instantly be shifted between location 1 and location 2. Thus, the DR scheme could be locally conducted within each zone with cheaper reserve from FR Unit, as indicated by flow 2 in Figure 6; then, the total cost will be reduced. Here, we assume the power network could afford to shift through virtual power lines. Figure 7 shows how the actual current flow on the network changes when the workloads shift between IDC1 and IDC2 through a virtual power line. The negative sign of the x-axis in Figure 7 denotes the workloads migrating from IDC2 to IDC1 and vice versa. As we discussed, the power change through virtual power lines could affect the actual power plow in the entire network. While the effects from virtual power lines on the actual power plow are complex and nonlinear due to the power loss, feasibility limits [3], renewables, network topology, etc. Figure 7 shows different safe regions for workload migration without causing operational limit violations when considering different lines. Such an investigation into the safe region and effects of virtual power transfer will be more complex and crucial when the number of IDCs continues to rise.

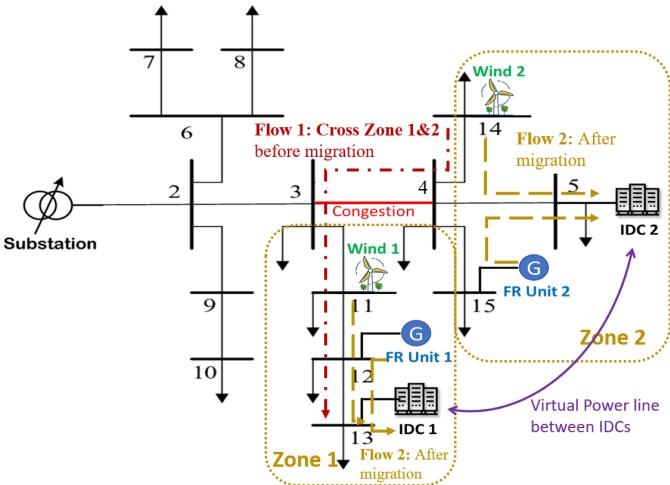

**Figure 6.** Illustration of congestion alleviation through network analysis.

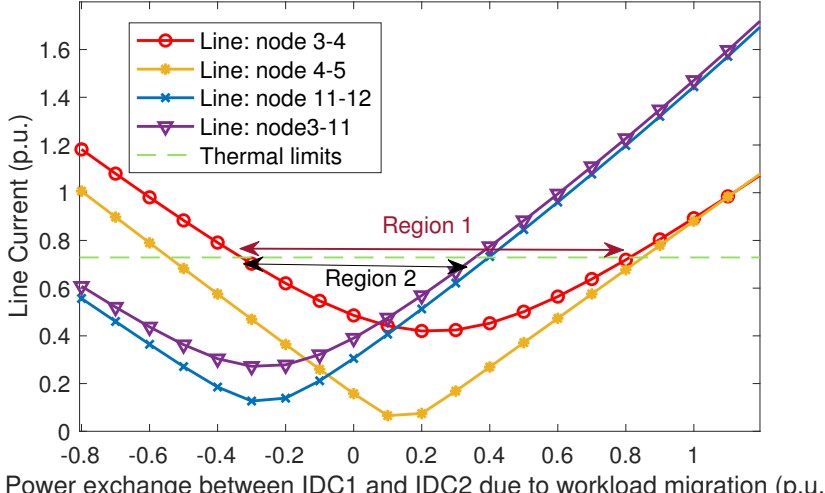

**Figure 7.** The changing trend of actual current flow in the network caused by the power shifting through virtual power line.

Simulation results in Figures 8 and 9 also prove that IDCs' workload migration contributes to both the operational benefits of power systems and the economic benefits of

IDCs. For demonstration, IDC2 is assumed to have heavier loads, which could cause congestion. As we want to shift the demand of IDC 2 to IDC 1, power systems will provide a higher DR price for IDC2. In this case study, IDC1's DR price IS 35 \$/MW while IDC2 is 43 \$/MW, which indicates a higher total revenue-joining DR scheme if IDC2 could provide more DR resources. Figure 8 shows that the more that workloads shift from IDC2 to IDC1, the more DR resources IDC2 can provide, and the more DR revenue IDCs can have. The downward curve of IDC2 workload average value indicates that the high-value workloads have higher priorities regarding migration and completion; thus, the low-value workloads are maintained in IDC2 with the possibility of being terminated to provide DR resourcesm and vice versa for the curve trends in IDC1. Figure 9 shows the operational costs of power systems during IDC workload migration. It shows that, although the DR purchase cost is slightly increased, the reserve cost and the total cost are reduced because more DR resources are available. Moreover, workload migration among IDCs can effectively reduce the uneven power demands and resource distribution, thus contributing to transmission congestion. Note that the congestion limit on lines 3–4 are included as a hard constraint in the optimization problem to ensure the power flow on lines 3–4 stays below the constraints; otherwise, a penalty will be applied that incurs higher operational costs. In the case study, the power flow on lines 3–4 maintains its maximum value without causing an extra penalty.

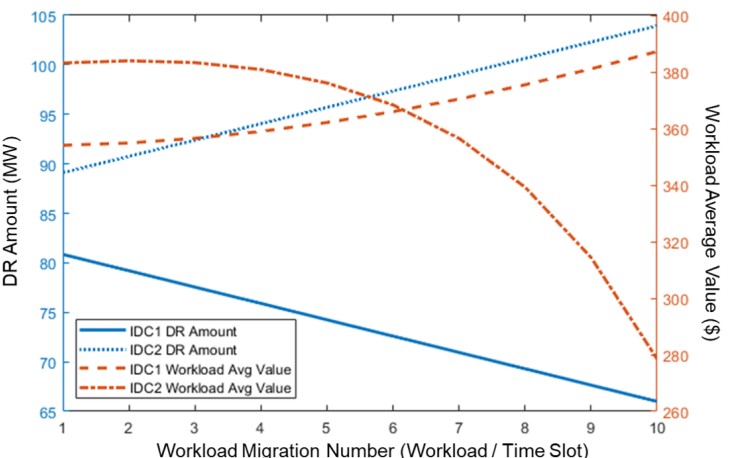

**Figure 8.** Effect of IDC Workload Migration to Provided DR Amount.

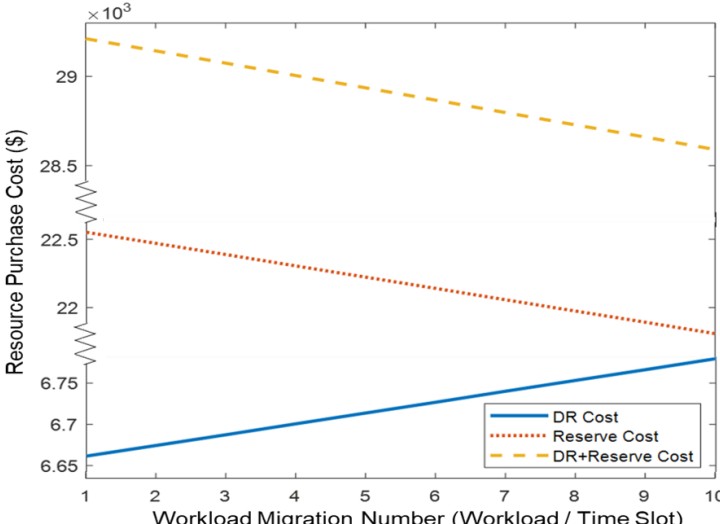

**Figure 9.** Effect of IDC Workload Migration to Resource Purchase Cost.

### 4.3. Influence of IDC Operation Uncertainties on DR Resource Purchase

As discussed, one of the main advantages of GPR is that the uncertainties in IDCs are reflected by the variance in the DR price–amount curve.

When there are sufficient sample data and no uncertainties in IDCs, renewables or price, GPR will be a curve, as indicated by the black curve in Figure 10, describing how the provided DR amount changes with the DR price with no variation. The accuracy of the black curve is decided by the available sample points [19]. When uncertainties in IDCs, renewables and price are considered, GPR predicts a variate range within a given confidence level, wherein the DR price–amount curve varies, as indicated by the grey area in Figure 10.

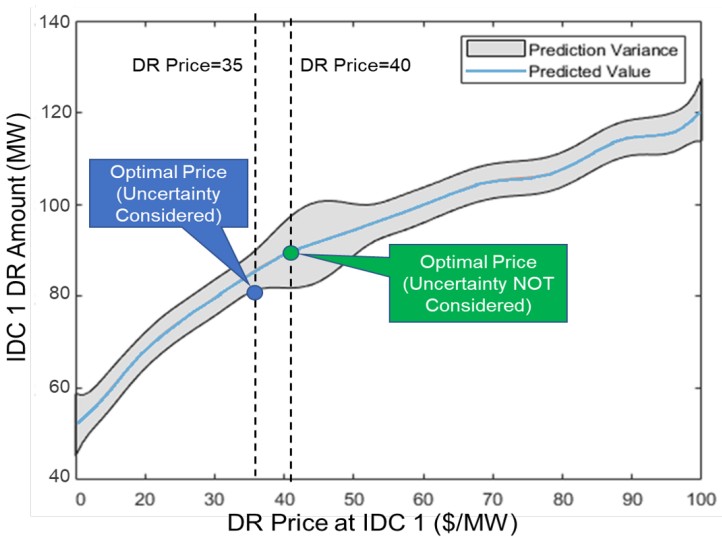

**Figure 10.** Optimal DR Price and DR Prediction Uncertainties.

Therefore, a robust DR price–amount curve should be the lower edge of the grey area, which indicates the lowest DR amount in most possible scenarios (95% in this paper) with a given DR price. From Figure 11, we can see that considering IDC uncertainties will change the DR price setting in the power system operation, and further change the purchased DR amount, as shown in Figure 10.

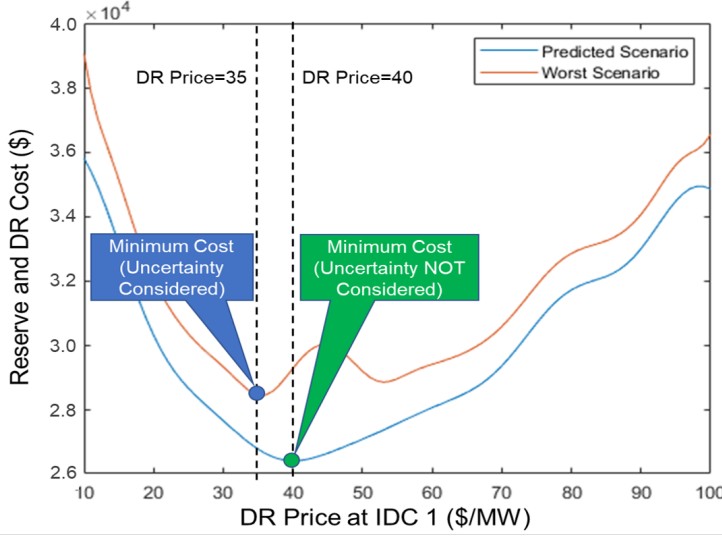

**Figure 11.** Operational Cost With/Without Considering DR Prediction Uncertainties.

### 4.4. Price–Amount Curve Construction and Analysis

Figures 12 and 13 illustrate the level of IDCs involved in demand response when power systems provide different pricestfor IDCs. One thousand samples are generated for each figure using the GUROBI solver. The IDC data for sampling follow the classic IDC model in reference [26] to ensure the IDC operation data is close to reality. Uncertainties are applied to IDCs workloads, renewable DG outputs, and electricity prices by variables following probabilistic distribution. In Figures 12 and 13, the x-axis and y-axis represent the DR price for each IDC, and the z-axis is the DR amount. As seen in each figure, the curve describes the changes in DR amount, which increase with DR price while IEEE 118 has a higher and smoother DR curve due to the robustness and larger size of the network (only two IDCs are allocated in the system to demonstrate the concept of the curve). These curves can be easily constructed with the proposed method, using historical data to facilitate operational demand-side management.

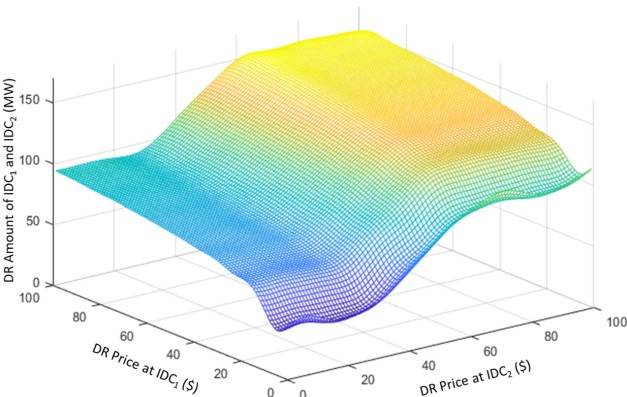

**Figure 12.** DR price–amount curve of multiple IDCs of IEEE 15-bus system.

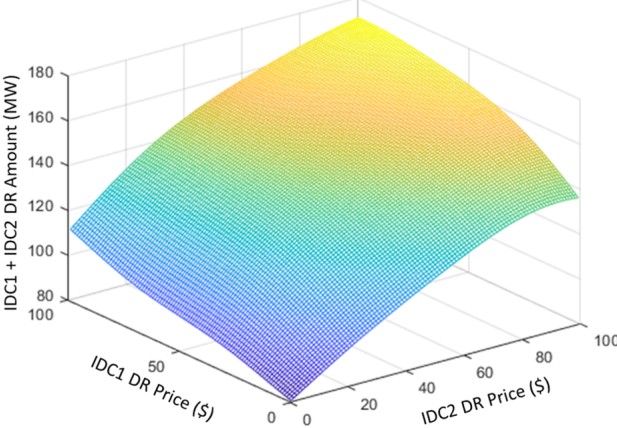

**Figure 13.** DR Price–Amount Curve of Multiple IDCs of IEEE 118-bus system.

### 4.5. Accuracy and Computational Efficiency Comparison

The proposed algorithm is a computationally efficient bi-level co-optimization model aiming to motivate IDCs to provide their spatial–temporal flexible loads as demand response resources to power systems. GPR approximates the function between DR quantity provided by IDCs and its price, while the power demand migration and operational uncertainties in IDCs are accurately described. The performance of GPR is affected by the selection of kernel functions. The kernel function reflects the distance between two variable points and is also called the covariance function. There are multiple methods to select the appropriate kernel for finite-dimension data [22]. Table 3 shows the accuracy of DR price–amount curve estimation with different kernel functionals and different ML methods. Default models in Matlab are used for the neural network regression model and the regres-

sion tree model without model specifications. From Table 3, it can be seen that, except for the linear kernel, the accuracy of GPR is below 4%, which is similar to the performance of the neural network regression and the regression tree. The linear kernel shows 13.6% errors due to the approximated linear function having low computational complexity and low accuracy to reflect the input–output relationship. The time needed for GPR is less than 1 s compared to 10 s of the neural network regression and 2 s of the regression tree, which shows that GPR is not costly. Note that the advantages of the neural network and the decision tree could be more obvious following the increase in the data size. While in this work, GPR is selected because of its advantage in acquiring the explicit form of functions, which can be easily embedded into the optimization problem, whereas neural networks or regression tree methods do not have such advantages.

**Table 3.** Comparison of the accuracy of the regressed DR price–amount curve using different methods.

| Method | Error | | Consumed | Method | Error | | Consumed |
|---|---|---|---|---|---|---|---|
| Kernel Function | Within Sample | Out of Sample | Time | Kernel Function | Within Sample | Out of Sample | Time |
| SE | 3.7% | 4.5% | <1 s | Matern 3/2 | 3.7% | 4.5% | <1 s |
| Exponential | 3.8% | 4.6% | <1 s | Matern 5/2 | 3.7% | 4.5% | <1 s |
| Linear | 13.6% | 15.8% | <1 s | Rational Quadratic | 3.7% | 4.6% | <1 s |
| Regression Tree | 3.7% | 4.5% | 2 s | Neural Network | 0% | 6.8% | 10 s |

The proposed methods can also be used in other larger-size networks such as IEEE 33, 118, and 123-bus systems with similar results. Compared to conventional two-layer optimization problems, the proposed methods can largely reduce the complexity of the co-optimization problem by minimizing the variables inherited from the lower-layer models [19], such that the total computational cost is mostly decided by the upper-layer model. In the tests on the 33-bus system, for a model with 260 variables in the upper layer, and 201 variables in the lower layer, both the proposed method and conventional methods can find the solution in 1 s, while the similar tests on the 123-bus system with a 20-fold increase in the number of variables, the conventional methods could fail to find a solution but the proposed methods still can find a local optimum in 177 s. In other words, the more complex the lower-layer model, the more advantageous the proposed methods. This point is significant for the co-optimization problem between different domains as the complexity of IDCs may not be less than the power systems following the increment of IDC deployment.

## 5. Conclusions

This paper introduces a robust bi-level co-optimization model that promotes the active participation of IDCs in demand response programs. The model leverages virtual power lines and Gaussian process regression to optimize resource allocations while respecting information barriers and accounting for uncertainties. By constructing data-based price–amount curves, the communication between power systems and IDCs is facilitated, ensuring computational efficiency and privacy protection. The proposed model and algorithm are validated using modified IEEE test systems, demonstrating their effectiveness. Overall, this research enhances collaboration between power systems and IDCs, leading to a more efficient and sustainable energy ecosystem. In the future, the power transfer on the virtual power grid and the actual power flow will be investigated to assist in the co-optimization of power systems and IDCs.

**Author Contributions:** Conceptualization, Y.L., Y.W. and H.D.N.; methodology, Y.L., Y.W. and H.D.N.; software, Y.L.; formal analysis, Y.W. and Y.L.; investigation, Y.W.; resources, R.L.T.L.; writing—original draft preparation, Y.W. and Y.L.; writing—review and editing, Y.W.; supervision, H.D.N. All authors have read and agreed to the published version of the manuscript.

**Funding:** This research is supported by NTU SUG, MOE AcRF TIER 1 RG60/22, EMA & NRF EMA-EP004-EKJGC-0003, NRF DERMS for Energy Grid 2.0, and Intra-CREATE Seed Fund NRF2022-ITS010-0005.

**Conflicts of Interest:** The authors declare no conflict of interest.

## Nomenclature

**Indices and Sets**

| | |
|---|---|
| $i, j$ | Index of nodes |
| $idc$ | Index of IDCs |
| $wl$ | Index of workloads in IDCs |
| $t, t'$ | Index of periods |
| $\mathcal{B}$ | Set of all nodes |
| $\mathcal{B}^{idc}$ | Set of IDC nodes |

**Parameters**

*Power System Operation*

| | |
|---|---|
| $PD^0$ | Power demand |
| $PD^{nomi}$ | Nominal power demand |
| $PG$ | Power generation, renewable units (uncertain) |
| $PG^0$ | Scheduled power generation, predicted value for renewable units |
| $\underline{PG}, \overline{PG}$ | Minimum and maximum power generation |
| $\underline{\theta}, \overline{\theta}$ | Minimum and maximum allowed voltage angle |
| $P^{Line}$ | Power line transmission capacity |
| $B$ | Susceptance between two nodes |
| $r^{up}$ | Unit upward ramping rate |
| $r^{down}$ | Unit downward ramping rate |
| $a^G, b^G$ | Generation operational cost coefficient |
| $a^R, b^R$ | Generation reserve operational cost coefficient |

*Internet Data Center Operation*

| | |
|---|---|
| $p^{WL}$ | Price of workload |
| $\underline{T}, \overline{T}$ | Release time and deadline of workloads |
| $v^l$ | Workload location indicator (binary, 1 = located in the IDC, 0 = otherwise) |
| $S^{req}$ | Required server resource amount of workloads |
| $S^{in}$ | Required server resource amount of incoming interactive computing tasks (uncertain) |
| $up^{max}$ | Maximum allowed uploaded workload number |
| $S^{cap}$ | IDC Server resource capacity |
| $PUE$ | IDC Power Usage Effectiveness |
| $\rho$ | IDC power consumption efficiency |
| $PG^{IDC}$ | IDC power generation (uncertain) |
| $E^{In}$ | IDC ESS initial energy level |
| $E^{Max}$ | IDC ESS maximum energy level |
| $p^E$ | Electricity price predicted by IDCs (uncertain) |
| $M$ | A very large number |
| $m$ | A small large number |

**Variables**

*Power System Operation*

| | |
|---|---|
| $C^G$ | Power generation operational cost |
| $C^{DR}$ | Demand response purchase cost |
| $C^R$ | Generation reserve operational cost |
| $PG$ | Power generation, conventional units (controllable) |
| $PG^0$ | Scheduled power generation, determined for conventional units |

| | |
|---|---|
| $PD$ | Actual power demand |
| $DR$ | Demand response amount |
| $\theta$ | Voltage angle |
| $R^{up}$ | Upward generation reserve |
| $R^{down}$ | Downward generation reserve |
| $p^{DR}$ | Price of Demand Response |
| *Internet Data Center Operation* | |
| $v^t$ | Workload termination indicator (binary, 1 = terminated, 0 = otherwise) |
| $v^c$ | Workload completion indicator (binary, 1 = completed in certain IDC, 0 = otherwise) |
| $v^u$ | Workload upload indicator (binary, 1 = uploaded to cloud, 0 = otherwise) |
| $S$ | Server resource allocated to workloads |
| $S^{usage}$ | Total server resource usage amount |
| $PD^{IDC}$ | IDC Power demand |
| $DR^{IDC}$ | Demand response provided by IDC |
| $PD^{IT}$ | Power demand of IT equipment in IDCs |
| $p^{ESS}$ | ESS charging/discharging power in IDCs (positive if charging, negative if discharging) |
| $E^{ESS}$ | ESS energy level in IDCs |
| **Functions** | |
| $C^G$ | Cost of generation |

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
