# Peer review of "Distributed Energy Resource Exploitation through Co-Optimization of Power System and Data Centers with Uncertainties during Demand Response"

_sustainability, doi:10.3390/su151410995_

Round 1

Reviewer 1 Report

Kindly check below comments/suggestion for your submission. 

1.     Introduction section can be improved. Please avoid the lumped references.

2.     In page no:6, Authors have insisted that interactive and flexible loads are considered in this work. How the scheduling of these two kinds of loads are differentiated? Did interactive loads also are considered for DR?

3.     Authors have mentioned that the workload migration among the IDCs reduces transmission congestion. Kindly include the figure / table to show the impact of the proposed work in congestion management for different timeslots.

4.     Authors have not mentioned the type of DR program implemented in the proposed work clearly. From line no: 106, it seems the DR is related to load curtailment. In that case, load curtailment will affect the consumer comfort. Kindly clarify the details of DR program considered and its impact on consumers.

5.     Authors also have discussed about the server allotment in page no:6. Equation 23 included the interactive load and flexible load of the respective IDC. How it will allocate during the migration of workload from other IDCs?

Author Response

Please see the updated PDF file. Thank you for your time and efforts.

Reviewer 2 Report

There is an article titled "A Gaussian Process-based Price-Amount Curve Construction for Demand Response Provided by Internet Data Centers" with the same premises and results from 2020 by two of the authors.

Author Response

(The authors gave the same response as above.)

Reviewer 3 Report

This interesting paper presents a computationally efficient bilevel co-optimization model for motivating IDCs to provide their spatial-temporal flexible loads as demand response resources to power systems. Following are the comments from the reviewer:

1.      It is recommended that the variables be explained in an individual section.

2.    Please explain why the error of Linear method in Table 2 is notably higher than that in other methods?

3.      It should be illustrated that how the blue curve in Fig.8 generated?

4.      The Procedures of DR Price-Amount Curve Construction with GPR shown in section 3.2 would be more readable if it was illustrated step by step (listed as Step 1, Step 2, …)

5.      The model has been verified in a modified IEEE 15-bus system, is it still applicable in a large-scale system? Also, the computation efficiency should be mentioned.

6.      Please correct the sentence structure, grammar, and typos.

Author Response

(The authors gave the same response as above.)

Reviewer 4 Report

This paper proposes a computationally efficient bilevel co-optimization model for motivating IDCs to provide their spatial-temporal flexible loads as demand response resources to power systems. With the proposed model, the optimal DR purchase scheme can be found to minimize the operational cost of power systems and IDCs with the consideration of uncertainties in the two systems, so that a win-win solution can be achieved for the both side.

The main topic of the study has a high degree of relevance.

The topic of optimizing the use of distributed generation facilities in the joint work of data centers and energy systems has a high degree of originality. A limited number of works are devoted to the same subject.

The presented article does not need improvements in terms of methodology.

The conclusions are quite consistent with the arguments presented.

In the presented article, links are appropriate.

The "Figure 5" does not indicate the rated voltage class of the microgrid in question.

Author Response

Please see the updated PDF file. Thank you for your time and support.

Reviewer 5 Report

This paper proposes a computationally efficient bilevel co-optimization model for motivating IDCs to provide their spatial-temporal flexible loads as demand response resources to power systems. The paper is  very interesting, with strong theory, clear logic and sufficient experiments and relevant. Nevertheless, the paper should undergo some minor review modifications before acceptance.

1.   Abstract Introduction should be revised to highlight the contribution of this paper.

2.   In the third part, the paper lacks the analysis of correctness, efficiency, rationality and limitation of the proposed co-optimization model, and suggests appropriate addition.

3.  In formula 40 to 43, the "objective"  and "Subject to"  need to be clearly written. It is not recommended to replace the previous formula number.

4.  In the fourth part, the explanation of IDC1 and IDC2 should be added, and the analysis of the results shown in Figure 6 should be added.

5.  I have not seen the results of the proposed method compared to other methods. How to prove the advancement and practicability of the method?

Author Response

(The authors gave the same response as above.)

Reviewer 6 Report

In this paper focus in better exploiting distributed energy resources by means of changing the conventional operation of data center to behave as demand response. Some models and the concept pf virtual transmission line are proposed. Various cases of study are carried out using the IEEE microgrid system.

 How “co-optimization” should be interpreted? …as a separated optimization of various individual systems?  Or as an integrated single optimization of various systems put together? …or?

The Authors state, “This large number will undoubtedly affect future system operations” please clarify, what this mean?

A microgrid does not represent a power system (generation and transmission). I have this discrepancy regarding the point of view of the Authors. The disagreement is because the microgrid is not governed by the dynamics of large synchronous generators and generally does not need auxiliary services.

The critical review of the state-of-the-art is very superficial. no pros and cons are discussed or valuated.

The authors do not discuss the problem statement. They just go directly to their proposal with no comparisons.

The Authors state, “Thus, a systematic solution to the aforementioned challenges is still missing with respecting the privacy of IDCs as well as being computationally effective.” is this the problem statement? Why is no other technical procedure described?

In general, the manuscript is quite wordy. Many text lines, paragraphs are not necessary to explain or detail the proposal. Example of this is “Following the rapid development of IDCs with increasing power demand, the high ……. and termination, etc.”

The Authors state, “The operation of each system will affect some key factors of the other ones.” specifically, which factors are these?

I think the concept of virtual transmission line is very good, however Author does not highlight this. Instead, Authors focus mainly in their models and algorithms.

The section “Interaction Between Power Systems and IDCs” is not well written. There are various information gaps.

From section 2.2 ahead, the set of equations cannot no be well understood because the reader must imagine (guessing) the meaning of parameters/variables.

The manuscript needs, in general, a profound review of the writing….this is language, edition, grammar, style, typos. Also, the manuscript is too wordy.

Author Response

(The authors gave the same response as above.)

Round 2

Reviewer 1 Report

The author response letter is good. The revised article is ready for publication. 

Author Response

The authors would like to thank the reviewer for your time and acknowledgment.

Reviewer 2 Report

In Table 3 you are presenting a comparison to other methods, but there is no description about them, did you implement those methods, or did you take them from the literature? There are methods, like a neural network that its success will depend on the data selected for its inputs and outputs, as well as the structure used. And in your article, there are no references to them. I recommend you address this last issue

Author Response

Please see the attached file for the response letter. Thank you for your time.

Reviewer 6 Report

I gladly find numerous good amendments in this version of the paper. The additions and corrections has resulted in more clear explanations.

Author Response

The authors would like to thank the reviewer for your time and acknowledgment. Your valuable comments helped us to improve this work.